# Study on the Hot Deformation Characterization of Borated Stainless Steel by Hot Isostatic Pressing

**DOI:** 10.3390/ma14237110

**Published:** 2021-11-23

**Authors:** Yanbin Pei, Xuanhui Qu, Qilu Ge, Tiejun Wang

**Affiliations:** 1Graduate School, Central Iron and Steel Research Institute, Beijing 100081, China; peiyanbin@atmcn.com; 2Institute for Advanced Materials and Technology, University of Science and Technology Beijing, Beijing 100083, China; 3Antai-Heyuan Nuclear Energy Technology and Materials Co., Beijing 100094, China; 4Advanced Technology and Materials Co., Ltd., Beijing 100081, China; wangtj@atmcn.com

**Keywords:** borated stainless steel, hot isostatic pressing, hot compression test, dynamic recrystallization, hot deformation characterization, critical stress, critical strain

## Abstract

Borated stainless steel (BSS) specimens have a boron content of 1.86 wt%, and are prepared by hot isostatic pressing (HIP) conducted at different temperatures, ranging from 1000 to 1100 °C and a constant true strain rate (0.01, 0.1, 1 and 10 s^−1^). These tests, with observations and microstructural analysis, have achieved the hot deformation characteristics and mechanisms of BSS. In this research, the activation energy (*Q*) and Zener–Hollomon parameter (*Z*) were contrasted against the flow curves: *Q* = 442.35 kJ/mol. The critical conditions associated with the initiation of dynamic recrystallization (DRX) for BSS were precisely calculated based on the function between the strain hardening rate with the flow stress: at different temperatures from 1000 to 1100 °C: the critical stresses were 146.69–254.77 MPa and the critical strains were 0.022–0.044. The facts show that the boron-containing phase of BSS prevented the onset of DRX, despite the saturated boron in the austenite initiated DRX. The microstructural analysis showed that hot deformation promoted the generation of borides, which differed from the initial microstructure of HIP. The inhomogeneous distribution of elements in the boron-containing phase was caused by hot compression.

## 1. Introduction

Borated stainless steels (BSS) containing more than 0.1 wt% boron, which are capable of absorbing thermal neutrons due to the high neutron cross-section of isotope ^10^B [1], are widely used within the nuclear fuel cycle field. BBS are used, for example, in spent fuel storage racks [2], which need a neutron absorber sheet or a square tube. BSS fabricated by billet casting [3] or powder metallurgy (PM) [4] need to be heat deformed into the above dimensions. PM is the best choice for those wishing to prepare a homogenous microstructure, and high-performance high-boron stainless steels. The process of hot isostatic pressing (HIP) of metal powders plays an important role in powder metallurgy [5,6,7]. HIP is a process used to increase the density of powders in a furnace at high pressure (100–200 MPa) and at high temperatures for BSS [4].

The material will be loaded with a range of stress, strain and temperature conditions during the deformation process. By testing the stress-strain curves at room [8] or high temperatures [9], key parameters can be obtained and the constitutive equations can be established. Various pieces of testing equipment used under various working conditions, for instance high temperature or high stress for different manufacturing processes, have been developed in recent years [10].

A mechanical and microstructural understanding of the dynamic processes that evolve during hot deformation, at different temperatures and strain levels, is important if one is to assess the hot workability of materials [11,12]. Dynamic recrystallization (DRX), which happens during the hot deformation process, is a key mechanism regarding microstructure and property control [12,13]. The generation and growth of the precipitation in austenitic steels during heat deformation has had a significant impact on its properties [14]. The precipitation of borides in BSS is M_2_B [15], therefore the DRX of the austenitic matrix phase, the behaviors of the boride and the diffusion of boron are critical for the hot deformation of BSS. In the past, the hot deformation behaviors of 304 stainless steel has been studied. For instance, Marchattiwar et al. studied the DRX behavior in hot deformation at 900–1200 °C for 304 austenitic stainless steel, over a strain rate ranging from 0.002 to 0.1 s^−^^1^, by conducting a hot compression test. The activation energy (Q) was calculated as *Q* = 475 kJ mol^−^^1^, by the regression analysis for a conventional hyperbolic sine equation [16]. The study of the effect of the initial grain size on the DRX process shows that by increasing the grain size, the range of stress becomes tighter and the characteristic peak of the DRX also tends to disappear [17]. The effect of twinning on the DRX process was investigated by Wang et al. Twinning developed the separation of the bulged region for DRX nucleation and made the extension of the DRX area [18]. The extensive research on the effect of 304 stainless steel grains and microstructure on the DRX process provides a good reference for the study of boron addition in stainless steel.

Low amounts of boron (tens to hundreds ppm) are added to steel to improve hardenability and enhance steel properties [19]. During hot deformation, low boron can reduce the hot deformation activation energy [20,21,22,23]. Boron, which segregates in austenite grain boundaries, will delay mobility and kinetics for the recrystallization of hot working, as boron drags solute atoms [24]. Gao et al. found that as the boron content increased, the flow stress decreased at low strain conditions, however, the austenite grain increased for boron microalloyed steel with three boron contents, 20, 40 and 60 ppm, at hot deformation test temperatures of 900 to 1100 °C and strain rates of 0.1 to 10 s^−1^ [20]. Mejía et al. found that the critical conditions for DRX initiating were dependent on the strain rate and the temperature. Notably, both the critical stress and critical strain decreased with an increased content of boron for microalloyed steel, with four different proportions of boron (29, 49, 62 and 105 ppm) with hot-compression tests at temperatures ranging from 950 to 1100 °C and three strain rates of 0.001, 0.01 and 0.1 s^−1^ [25].

Compared to low boron, the DRX of the austenitic matrix was accelerated remarkably in the presence of boride particles for AISI 304 stainless steel containing 1% boron by a prepared cast with a hot torsion test, which was contributed to by a higher overall strain energy stored in the austenite, and the superposition of both local strain accumulations [26]. In summary, both the solution of boron in austenite and the boride generated at the grain boundaries had an effect on DRX behavior.

BSS part dimensions are generally sheet or tube, so the alloys are acquired using heat deformation processing. The hot deformation characteristics of BSS by cast have been studied in a large number of papers [26,27,28,29], however there are only a few studies of BSS by HIP. Since the boron-containing phases, elemental distributions, etc., within the two processes differ [3,4], the study of the behavior of BSS by HIP during hot compression, especially Grade 304B7 according to ASTM A887, which is widely used in nuclear spent fuel racks, is essential in order to guide future research. The boride (Fe, Cr) _2_B in BSS and the austenite are heated at temperatures ranging from 1150–1225 °C in order to generate the eutectic melting point [30], and when the temperature is below 900 °C, the BSS in the deformation process may develop defects such as cracking, therefore the appropriate hot deformation temperature is 1000–1150 °C [31]. The hot deformation process of BSS may involve hot rolling with a lower strain, or hot extrusion with a higher strain, hence in this study the behavior of BSS containing 1.86 wt% boron was investigated at various temperatures ranging from 1000 to 1100 °C, and under strain rates from 0.1 to 10 s^−1^ by hot compression. After obtaining the hot compression stress-strain curves, various calculations were performed to obtain the characteristic parameters of hot deformation, such as the activation energy of hot deformation, critical stress, critical strain, etc. The observation of the microstructure and elemental distributions was used to analyze the differences in the DRX characteristics of BSS by HIP compared to those of low content boron steels (such as tens of ppm) and BSS by cast.

## 2. Materials and Methods

BSS alloy with 1.86 wt% boron, which was in accordance with Grade 304B7 of Standard ASTM A887, was fabricated via HIP at 1100 °C. Table 1 shows the chemical composition of the BSS. The density of the alloy was 7.62 g/cm^3^ according to ASTM B311. BSS by HIP consisted of two phases: the matrix austenitic phase and the boron-containing phase. The room temperature mechanical properties were: tensile strength of 740 MPa, yield strength of 370 MPa and an elongation of 13%.

Cylindrical specimens with dimensions of φ8 × 12 mm were machined after being taken out of the billet. Manufactured by Dynamic Systems Inc., the Gleeble-3500 thermo-mechanical simulator was used for this test. Three different deformation temperatures (1000, 1050, and 1100 °C) and four strain rates (0.001, 0.1, 1, and 10 s^−1^) were carried out in hot compression tests, with the true strain reaching 0.8. Regarding the heat treatment of the alloys, the specimens were heated to temperatures of 1000, 1050 and 1110 °C, respectively, taking four minutes to compress, before they were then immediately quenched in water.

After the hot compression tests, the specimens were divided equally in half along the radial direction. The separated faces were polished and analyzed. Microstructures were observed by a Nova nanoSEM 450 field-emission scanning electron microscope (FE-SEM) with secondary electron (SE) detection, produced by the FEI company and an X-Max50 energy-dispersive spectroscopy (EDS) made in Oxford Instruments analysis.

## 3. Results

### 3.1. True Stress-Strain Curves

Figure 1 shows that the true stress-strain (*σ*-*ε*) curves of the BSS that were deformed at discrete temperatures of 1000, 1050 and 1100 °C with different strain rates. One can see that both the deformation temperatures and the strain rates had a significant effect on the deformation characteristics of the BSS. As such, Figure 1 shows that the flow stress of the tested BSS raises when the temperature drops and the strain rate increases. The stress–strain curves demonstrate the distinct classification of mechanisms about dynamic recovery (DRV) and DRX. Generally speaking, the occurrence of DRX shows the stress rises to the peak and softens toward the steady region, and DRV has no clear peak stresses for the flow curves. DRV is visible when the deformation was at 1000 and 1100 °C under the strain rate of 0.1 s^−1^. DRX occurred at other points during the deformation process other than those shown in the two curves above.

Under lower strain rates or high deformation temperatures, multi-peaked stress-strain curves may occur. The flow curve at 1050 °C and under the strain rate of 0.1 s^−1^ was the typical multi-peaked one. With a temperature of 1100 °C and strain rate of deformation of 0.1 s^−1^, there was a significant drop in the curve at an approximate strain value of 0.7, which then raised again; the similar trend was seen when the temperature reached 1050 °C and the strain value hit 0.1 and 1 s^−1^. There were some small fluctuations in the curve at 1050 °C and a strain rate under 0.1 s^−1^, probably due to the fact that at this temperature and strain, boron diffused faster than at low temperatures and had a longer diffusion time due to the low strain, and the boride precipitation generated after diffusion to the grain boundaries. The presence of boron generated DRX, yet the boride generation played the opposite role, hence the fluctuations. In fact, smaller fluctuations can be observed at lower deformation rates (0.1 and 0.01 s^−1^); this temperature (1050 °C) and strain may have achieved balance, increasing the curve fluctuations.

### 3.2. Constitutive Analysis for BSS

Constitutive analysis is very important in the investigation of the alloy deformation. Generally speaking, the effects of the temperatures and strain rates on the deformation behaviors can be represented by the Zener–Hollomon parameter (*Z*), which is a key parameter for hot deformation and represents the effect on deformation behaviors by both temperatures and strain rates. The equation of *Z* is an exponent-type one (Equation (1)).

The relation of the Z parameter and the stress is given by the hyperbolic sine law (Equation (2)) in the Arrhenius equation. The power law description of stress is suitable for relatively low stresses with *ασ* < 0.8, and for high stresses with *ασ* > 1.2, the exponential law can be used (Equation (3)) [32].
(1)$$Z=\dot{\varepsilon }\exp(Q/\mathrm{RT})$$
(2)$$\dot{\varepsilon }\;=\;A{\left[\sinh(\alpha \sigma )\right]}^{n}\;\exp(-Q/\mathrm{RT})\;\mathrm{for}\;\mathrm{all}\;\sigma$$
where
(3)$${\left[\sin h\left(\alpha \sigma \right)\right]}^{n}=\{\begin{array}{c} \sigma^{n1}\to \alpha \sigma <0.8\\ \exp\left(\beta \sigma \right)>1.2 \end{array}$$
in which R is the universal gas constant (*R* = 8.31 J·mol^−1^·K^−1^), $\dot{\varepsilon }$ is the strain rate (unit: s^−1^), *Q* is the activation energy of hot deformation (unit: kJ/mol), T is the absolute temperature (unit: K), *α*, *β*, *A*, *n*1 and *n* are the material constants, and *σ* is the flow stress (unit: MPa). Such characteristic stresses as the peak stress and the steady stress matching to the specific strain can be used to calculate these equations. The constitutive equations for BSS were set up by peak stress. The α in the Equation (2) can be calculated directly from *α* = *β*/*n*1, and the average slope of the lines (ln$\dot{\varepsilon }$-ln*σ*_p_ and ln$\dot{\varepsilon }$-*σ*_p_ plots) can conclude the value of *n*1 and *β*, respectively [33].

After taking natural logarithms on both sides of Equation (2), the result is:(4)$$\ln\dot{\varepsilon }\;=\;\ln A\;+\;n1\ln\sigma \;-\;(Q/\mathrm{RT})$$
(5)$$\ln\dot{\varepsilon }\;=\;\ln A\;+\;\beta \sigma \;-\;(Q/\mathrm{RT})$$

Equation (4) is for low (*ασ*_p_ < 0.8) stress levels and Equation (5) is for high levels (*ασ*_p_ > 1.2).

Figure 2 shows both the tested data and regression results of ln$\dot{\varepsilon }$-ln*σ*_p_ and ln$\dot{\varepsilon }$-*σ*_p_ plots for the BSS, and the average value of *n*1 and *β* at three temperatures can be computed: *n*1 = 6.2185 and *β* = 0.0356, then α = 0.0057.

Equation (6) can be obtained after making natural logarithms on both sides of Equation (2).
(6)$$\ln\dot{\varepsilon }\;=\;n\;\ln[\sinh(\alpha \sigma_{P})]\;+\;\ln A\;-\;Q/(\mathrm{RT})$$

If T is constant, the partial differentiation for Equation (6), Equation (7) can be yielded.
(7)$$n=\frac{\partial \;ln\dot{\varepsilon }}{\partial \;ln\left[\sinh\left(\alpha \sigma_{p}\right)\right]}$$

In accordance with the relationship curves of ln*ε* and ln[sinh(*ασ*_p_)] (Figure 3a), the average value of *n* can be estimated to be 4.5633 for the BSS.

If $\dot{\varepsilon }$ is constant, the partial differentiation of Equation (6), Equation (8) can be obtained.
(8)$$Q=Rn\frac{\partial ln\left[\sinh\left(\alpha \sigma_{p}\right)\right]}{\partial \left(1/T\right)}\;$$

In accordance with the relationship curves of ln[sinh(*ασ*_p_)] and T^−1^ (Figure 3b), which was one group of parallel lines, the average value of the slopes was 11,659. So the value of *Q* can be calculated to be 442.35 kJ/mol.

The equation below can be yielded, in accordance with Equations (1) and (2).


(9)
$$Z\;=\;A{\left[\sinh(\alpha \sigma_{p})\right]}^{n}$$


The corresponding curves of [sinh(*ασ*_p_)]*^n^* with *Z*, is shown in Figure 4, from which *A* can be calculated as 4.570 × 10^16^. The constitutive equations of the experimental steels are shown as follows:


(10)
$$Z\;=\;4.570\;\times \;10^{16}{\left[\sinh(0.03557\sigma_{p})\right]}^{6.2185}$$


### 3.3. Critical DRX Parameters

The true stress (*σ*) and strain hardening rate (*θ* = d*σ*/d*ε*) estimates exactly the characteristic value of the flow plot, and can been used to obtain the peak stress (*σ*_p_) and critical stress corresponding to the critical strain (*ε*_c_) in general. In the *θ*-*σ* plot, the peak stress corresponds to the points at which the plot crosses the zero from above, and the initiation DRX initiates when the marked minimum in the (−(d*θ*/d*σ*))-*σ* plot appears.

Figure 5 and Figure 6 show the *θ*-*σ* curves and the −(d*θ*/d*σ*)-*σ* curves for the BSS under strain rates of 10 and 1 s^−1^ at 1100, 1050, and 1000 °C. That the peak stress and critical stress markedly increased with decreasing temperatures was obvious. According to this method, the peak stress, critical stress, peak strain and critical strain can be precisely measured, as is shown in Table 2.

The obtained values for peak stress, peak strain, critical stress and strain are summarized in Table 2, and are compared with other similar steels in other studies. Low boron can reduce the hot deformation activation energy, critical stress, and critical strain [25]. However, boron is considered insoluble in Fe-Cr alloys, and the solubility value of between 0.004 mass% and 0.008 mass% has been found at 900 °C [34]. Therefore, if steel contents meet high content boron, boron will segregate at the grain boundaries and form borides [3,4]. Due to the presence of borides, the peak stresses in this study were higher than those of AISI 304 stainless steel [35] without boron, being more than 50% higher at the same temperature. The peak stresses of the BSS, with a content of 1.96 wt% boron [27], were also higher than in this study due to the increase in boron content and the amount of borides. As temperature and stress increased, boron diffusion accelerated and solubility in the steel increased, therefore the boron content in the austenitic matrix increased, promoting DRX; boron in the steel also boosted DRX, which was the reason why the critical strain of the BSS was below 0.05.

### 3.4. Metallographic Structure

Table 3 summarizes the contents of the different elements, such as B, Fe and Cr, for the SEM-EDS point scan analysis shown in Figure 7. Prepared by HIP, the BSS alloy contained two phases: the matrix austenite phase and the boron-containing phase containing borides [4]. In contrast to the microstructure before the hot compression test (Figure 7a), the following changes in the microstructure occurred after the hot compression test. Firstly, there were the light-coloured austenitic phase, the dark-coloured boron-containing phase and the black phase; the austenitic and boron-containing phases had diffusion zones. Secondly, there was the black phase, whose size was less than 1 micron and was generated at the grain boundaries; the compositional analysis (spectrum 4 and 10) indicated the presence of a large amount of boron. Thirdly, the black phase was generally connected to the boron-containing phase, indicating that the former was transformed by the latter during the hot compression test.

The black phase was absent for the BSS alloy after HIP or heat treatment [4], but was generated after the hot compression test, indicating that the distortion energy of hot compression activated the generation of the black phase. Under a low strain rate (0.01 s^−1^), the black phase contained high levels of boron, 34.7 wt% and 25.9 wt%, respectively, (spectrum 4 at 1000 °C, spectrum 10 at 1100 °C). Under a high strain rate (10 s^−1^), the black phase was observed to have no boron by EDS (spectrum 7 and 14), possibly due to the fact that EDS is not accurate for boron with a low atomic number. The diffusion time explained why the boron content under the strain rate of 0.01 s^−1^ was higher than that under the strain rate of 10 s^−1^.

The darker boron-containing phases (spectrum 1, 5, 8 and 11) contained similar amounts of Cr and Fe as during the HIP and heat treatment [4]. The boron-containing phase was the mixture of black and lighter colored ones; Cr and Fe were observed in high content, while Ni was in low content in the black region. The light-colored regions are the diffusion zones, containing more Cr (spectrum 2, 6 and 12) than the austenitic.

## 4. Discussion

One can observe from the distribution of boron in the steel, that there was a small amount of boron solution in the austenite matrix and in the grain boundaries to generate borides. First for the austenitic matrix, it is generally accepted that a low amount of boron will reduce the hot deformation activation energy, and more boron content will need less deformation for the onset of DRX [18]. The matrix phase of the BSS is the saturated solid solution of boron, so it requires less deformation for DRX to occur. Therefore, one can see that the critical strain of the alloy ranges from 0.022 to 0.044 s^−1^ and the peak strain ranges from 0.135 to 0.232 s^−1^, and these data are much lower than those data captured on 304 stainless steel with no boron, as is shown in Table 2.

Due to the low solubility of boron in steel [34], borides have been generated by the boron merging with Fe, Cr, etc., in the austenite grain boundary. The boron-containing phase or boride blocks the movement of dislocation, so the peak and critical stresses are higher than that of AISI 304 stainless steel [35], which has a similar composition to BSS. In summary, the boron phase raises the critical and peak stresses and the boron saturated in the matrix phase lowers the critical and peak strains.

The BSS prepared by HIP has several characteristics compared to the cast alloy: firstly, the second phase composition is different: the boron-containing phase of the BSS by HIP is the mixture of Cr-Fe solid solution and borides, while the cast BSS alloy is borides. Secondly, the distribution of the second boron-containing phase is not the same. Thirdly, the grains of BSS by HIP are finer than that of the cast. For as-cast BSS alloy, boride particles can be the particle-stimulated nucleation sites, yet only a small fraction of the new grains formed by the particle-stimulated nucleation [28]. The above factors, finer grains with more grain boundaries and the mixture of solid solution with boron phases deformed at lower stresses, cause boron of BSS by HIP to diffuse more easily than that of the cast. Therefore, the activation energy of hot deformation for BSS by HIP (Q = 442.35 kJ/mol) is lower than that of the cast alloy (620.32 kJ/mol) [27].

On the other hand, hot deformation boosts the growth of the boron-containing phase, which in Figure 7b–d have a grain size of approximately 5 µm, which is larger than the initial grain size (in Figure 7a: the grain size is below 3 µm). Secondly, a high content of borides (black phase in Figure 7b–d) are induced to be generated by hot deformation, at the lower strain rate (0.01 s^−1^). The boron content in different borides varies due to the diffusion time. Thirdly, hot deformation has contributed to the diffusion of elements other than boron. The boron-containing phase has dark and light colored regions, with the darker region having a high Cr content and the lighter region having the lower Cr content, however in the austenitic phase the Cr content is much lower compared with that in the boron-containing phase. There are diffusion zones of Cr, Fe and other elements between the boron containing phase and austenite. However, the main reason derives from the different binding forces of Cr, Fe and Ni [36], affecting the diffusion.

## 5. Conclusions

The objective of this paper was to obtain data on the hot deformation characteristics and mechanisms of the Grade 304B7 by HIP: the activation energy is concluded by true *σ*-*ε* curves and calculated: *Q* = 442.35 kJ/mol; the constitutive equations are obtained as follows: *Z =* 4.570 × 10^16^[sinh(0.03557*σ*_p_)]^6.2185^.Different perspectives were found and laid out: DRX of BSS by HIP initiates under low strain (critical strains: 0.022–0.044) and high stress (critical stresses: 146.69–254.77) with different results than that of normal stainless steels, low boron steels and cast BSS.Microstructure evolutions were observed differently to low boron steels and cast BSS: the hot compression boosts the generation of borides with high content boron at austenite and boron-containing phase grain boundaries.

## Figures and Tables

**Figure 1 materials-14-07110-f001:**
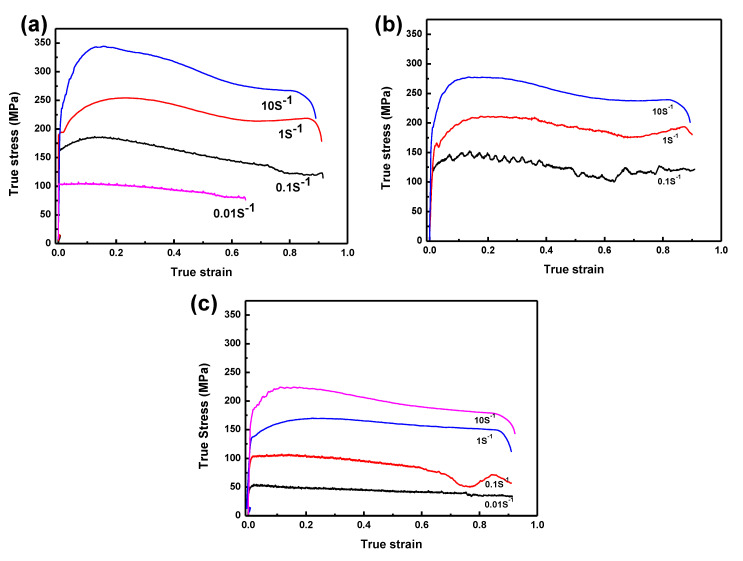
Stress-stain (*σ*-*ε*) curves for borated stainless steel (BSS) at different temperatures (**a**) 1000 °C; (**b**) 1050 °C; (**c**) 1100 °C and various strain rates.

**Figure 2 materials-14-07110-f002:**
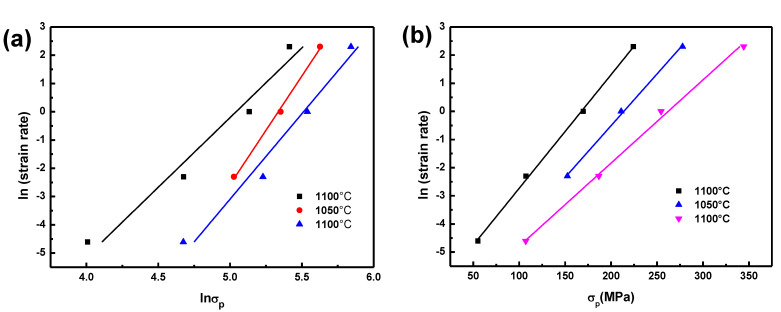
Relationship between ln$\dot{\varepsilon }$ and (**a**) ln*σ*_p_; (**b**) *σ*_p_.

**Figure 3 materials-14-07110-f003:**
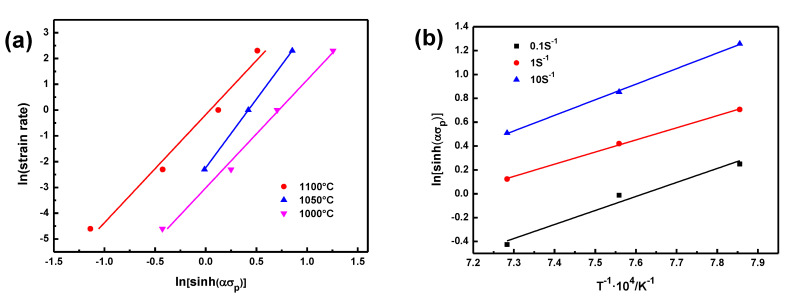
Relationship between: (**a**) ln[sinh(ασ_p_)] and ln$\dot{\;\varepsilon }$, (**b**) ln[sinh(*ασ*_p_)] and 1/T.

**Figure 4 materials-14-07110-f004:**
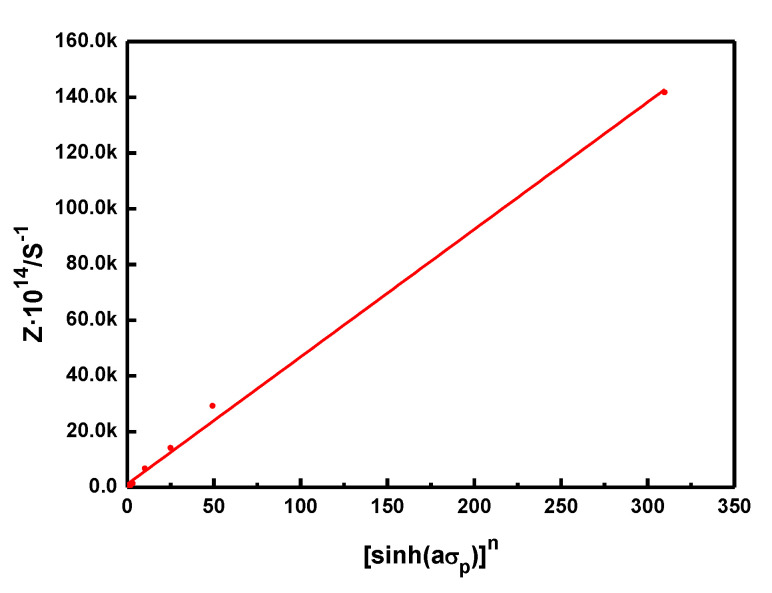
Relationship between *Z* and ln[sinh(*ασ*_p_)].

**Figure 5 materials-14-07110-f005:**
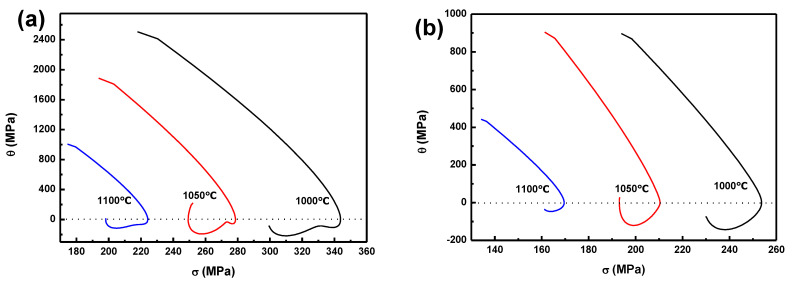
Curve of *θ*-*σ* 1 s^−^^1^ at three temperatures, under two strain rates (**a**) 10 s^−^^1^ and (**b**) 1s^−^^1^.

**Figure 6 materials-14-07110-f006:**
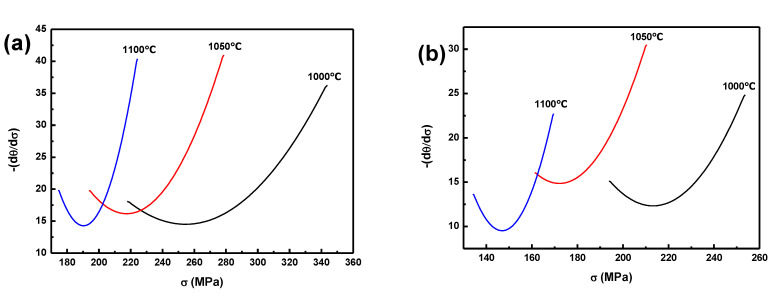
Curve of –(d*θ*/d*σ*)-*σ* at three temperatures, under two strain rates (**a**) 10 s^−^^1^ and (**b**) 1s^−^^1^.

**Figure 7 materials-14-07110-f007:**
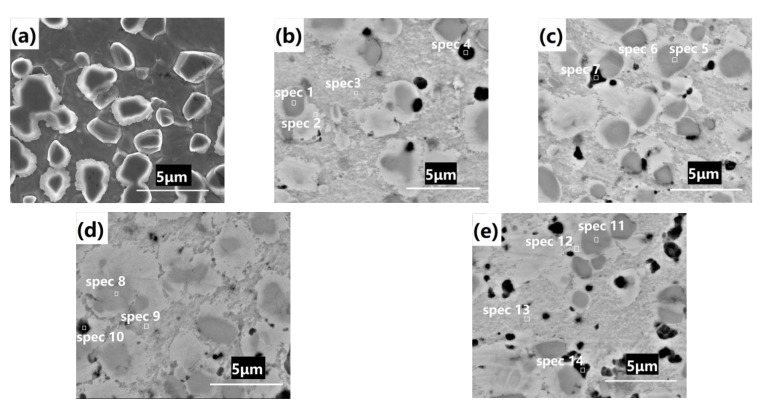
Microstructure of BSS (**a**) hot isostatic pressing; microstructure and EDS point analysis of BSS after hot compression test at different temperatures under different strain rates (**b**) 1000 °C and 0.01 s^−1^, (**c**) 1000 °C and 10 s^−1^, (**d**) 1100 °C and 0.01 s^−1^, (**e**) 1100 °C and 10 s^−1^.

**Table 1 materials-14-07110-t001:** Chemical composition of investigated alloys in wt%.

Element	B	C	Cr	Ni	Mn	Si	P	S
wt%	1.86	0.020	19.30	14.10	2.00	0.66	0.0070	0.0051

**Table 2 materials-14-07110-t002:** Peak stress, peak strain, critical stress and strain for 304B7 BSS and related steels.

Steel	Temperature (°C)	$\dot{\varepsilon }$ = 10 s^−1^			$\dot{\varepsilon }$= 1 s^−1^		
		*σ*_p_ (MPa)	*σ*_c_ (MPa)	*σ*_c_/*σ*_p_	*σ*_p_ (MPa)	*σ*_c_ (MPa)	*σ*_c_/*σ*_p_
304B7 (This study), Boron content: 1.86 wt%.	1000	344.36	254.77	0.74	254.35	212.65	0.83
	1050	277.75	216.95	0.78	211.03	172.69	0.82
	1100	224.33	190.61	0.85	169.80	146.69	0.86
		** *ε* _p_ **	** *ε* _c_ **	***ε*_c_/*ε*_p_**	** *ε* _p_ **	** *ε* _c_ **	***ε*_c_/*ε*_p_**
	1000	0.157	0.022	0.139	0.232	0.037	0.159
	1050	0.135	0.023	0.167	0.221	0.044	0.198
	1100	0.157	0.028	0.179	0.222	0.041	0.184
**Steel**	**Temperature (°C)**	***σ*_p_ (MPa)**	***σ*_c_ (MPa)**	** *σ* _c_ ** **/*σ*_p_**	***σ*_p_ (MPa)**	***σ*_c_ (MPa)**	** *σ* _c_ ** **/*σ*_p_**
304B7 [27], Boron content: 1.96 wt%.	1000	350	-	-	280	-	-
	1050	290	-	-	240	-	-
	1100	270	-	-	210	-	-
**Steel**	**Temperature (°C)**	$\dot{\varepsilon }$ **= 5 s^−1^**			$\dot{\varepsilon }$ **= 0.5 s^−1^**		
		***σ*_p_ (MPa)**	** *ε* _p_ **	***ε*_c_/*ε*_p_**	***σ*_p_ (MPa)**	** *ε* _p_ **	***ε*_c_/*ε*_p_**
AISI 304 [35]	1000	192	0.89	0.73	133	0.63	0.73
	1100	145	0.92	0.73	99	0.6	0.73

**Table 3 materials-14-07110-t003:** Elements content of EDS point scan analysis of BSSs after hot compression test (wt%).

	Spectrum	B	Fe	Cr	Ni	Mn	Si
Figure 7b	1	-	45.7	50.1	2.2	2.1	-
	2	-	62.4	21.1	13.3	2.2	1.0
	3	-	66.3	15.0	15.7	2.1	0.9
	4	34.7	44.1	11.6	9.4	-	0.2
Figure 7c	5	-	44.4	53.9	1.8	-	-
	6	-	62.7	20.7	14.0	1.9	0.7
	7	-	66.6	14.7	15.7	2.4	0.6
Figure 7d	8	-	45.1	49.5	2.6	2.8	-
	9	-	66.3	14.0	16.2	2.7	0.8
	10	23.3	49.7	12.3	12.1	2.2	0.4
Figure 7e	11	-	45.2	49.9	2.5	2.3	-
	12	-	63.2	18.2	15.1	2.2	0.9
	13	-	66.4	16.4	14.0	2.3	0.9
	14	-	60.2	26.3	10.5	2.1	0.8

## Data Availability

Data available in a publicly accessible repository.

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
