# Peer review of "Study on the Hot Deformation Characterization of Borated Stainless Steel by Hot Isostatic Pressing"

_materials, 2021, doi:10.3390/ma14237110_

Round 1
Reviewer 1 Report
The current manuscript investigates the characteristics of borated stainless steel after applying the hot isostatic press. There are major issues that should be considered as follows:
- The title should be adjusted, "hot compression test" is not suitable to describe the contents, Hot isostatic press should be included.
- The introduction section is very weak. A critical review should be conducted. In addition to clearly defining the problem statement and the outline of the current work.
- The materials and methods section lacks a lot of information about the material properties, sample preparation, the instruments used in the experimental work.
- For Figure 1, it is recommended to unify the vertical access scale to present a fair and obvious comparison through the tested regimes.
- The title of Table 2 needs to be revised.
- Figure 7, Image "d" should be the same scale as other images.
- The discussion lacks the justification of the obtained results.
- The conclusion section just summarizes the results. There is no novelty or significant contribution presented.
Author Response
Dear Review:
Thank you very much for conscientious, professional and responsible comments on our paper. We have revised the manuscript according to your kind advice and detailed suggestions. Enclosed please find the responses to the referees.
Best regards
Sincerely yours
Yanbin Pei

Reviewer 2 Report
The paper presented the interesting view and research about influence of the Boron addition to the stainless steels, and it's influence to to reduce the hot deformation activation energy. The structure of the article as well as it's contents are correct and give clearly explanation of the issue.
I have only a few advises for English spelling:
1) too many times is used word "distinct" - please to replace others words like: discrete, marked, obtained;
2) too many times is used word "promote" - please to replace others words like: boost, support, initiate;
3) line 104 not "rose" but "raise";
4) line 251 add "T" to he.
Author Response

(The authors gave the same response as above.)

Reviewer 3 Report
The paper “Study on the Behavior of Powder Metallurgy Borated Stainless Steel by Hot Compression Test” studies the behavior of BSS containing 1.86 wt% boron was investigated at different temperatures ranging 1000 to 1100 °C under strain rates from 0.1 to 10 s-1 by hot compression.. It is of interest and novelty, so I suggest considering it for publication after minor changes:
- A graphical abstract would add interest to catch the eye
- The abstract correctly states the activities carried out but should also include the novelty and objectives of the study. Please supplement it.
- To increase the number of references, more references to strain-stress related papers, can be introduced:
Bhujangrao, T.; Froustey, C.; Iriondo, E.; Veiga, F.; Darnis, P.; Mata, F.G. Review of Intermediate Strain Rate Testing Devices. Metals 2020, 10, 894. https://doi.org/10.3390/met10070894
Hundy BB, Green AP. A determination of plastic stress-strain relations. J Mech Phys Solids. 1954;3(1):16-21. https://doi.org/10.1016/0022-5096(54)90035-6.
- Can you explain the fluctuations presented in Figure 1 b) in the test performed at 0.1s-1?
- Conclusion number 2 is poorly worded and makes no sense.
Author Response

(The authors gave the same response as above.)

Reviewer 4 Report
Dear Authors
Congratulations on doing some interesting research and developing the paper.
Below are some comments that I believe should be introduced into the text.
- The purpose of the research and analysis should be clearly indicated.
- The range of temperature and constant true strain rate selected for the tests should be justified.
- Explain the abbreviation DRV.
- The conclusions should be described in more detail.
Author Response

(The authors gave the same response as above.)

Round 2
Reviewer 1 Report
The revised manuscript is significantly improved. a minor point for the conclusion section; the points of the conclusion should be focused and concise. The main contribution and novelty should be obviously presented.
Author Response
Dear Review:
Thank you very much for quick, conscientious, professional and responsible comments on our paper. We have revised the manuscript according to your kind advice and detailed suggestions. Enclosed please find the responses to the referees.
Best regards
Sincerely yours
Yanbin Pei
Please find the following Response to the comments of referees:
